# ORCA: ENHANCING ROLE-PLAYING ABILITIES OF LARGE LANGUAGE MODELS BY INTEGRATING PERSONALITY TRAITS

## ABSTRACT

Large language models has catalyzed the development of personalized dialogue systems, numerous role-playing conversational agents have emerged. While previous research predominantly focused on enhancing the model's capability to follow instructions by designing character profiles, neglecting the psychological factors that drive human conversations. In this paper, we propose Orca, a framework for data processing and training LLMs of custom characters by integrating personality traits. Orca comprises four stages: (1) Personality traits inferring, leverage LLMs to infer user's BigFive personality trait reports and scores. (2) Data Augment, simulate user's profile, background story, and psychological activities. (3) Dataset construction, personality-conditioned instruction prompting (PCIP) to stimulate LLMs. (4) Modeling and Training, personality-conditioned instruction tuning (PTIT and PSIT), using the generated data to enhance existing open-source LLMs. We introduce OrcaBench, the first benchmark for evaluating the quality of content generated by LLMs on social platforms across multiple scales. Our experiments demonstrate that our proposed model achieves superior performance on this benchmark, demonstrating its excellence and effectiveness in perceiving personality traits that significantly improve role-playing abilities.

## 1 INTRODUCTION

Building human-like conversation agents is a long-term challenge for AI researchers. The emergence of groundbreaking language models such as ChatGPT and GPT-4 (OpenAI, 2023), coupled with their intrinsic capacity for emergent in-context learning (ICL) (Brown et al., 2020) abilities and a three-stage reinforcement learning from human feedback (RLHF) (Ouyang et al., 2022) algorithm which have largely raised the capacity bar of existing AI systems.

LLMs have acquired a wealth of knowledge during their pre-training stage. ICL utilizes LLMs in a few-shot or zero-shot way that can instruct LLMs to understand the tasks in the form of natural language text. Therefore, personality-based responses have gained significant attention. Despite GPT-4 exhibit advanced role-playing capabilities because human-generated conversations are combined with the Instruct tuning dataset in a dialogue format for training, it is widely recognized that LLMs, suffer from a lack of consistent personality traits often failing to be engaging. This is the result of the existing LLMs are predominantly trained on general domains and lack specific optimization for personalized LLMs.

Recently, human social behavior is being changed by role-playing applications, such as Character.AI which has attracted a growing number of researchers to bridge the gap between the text and behavior of dialogue agents and humans (Team, 2023; Wang et al., 2024). Personality-based dialogue systems can be broadly categorized into two types. (1) Persona-based Dialogue, represented by the work in Zhang et al. (2018) where the manipulation of profile information is employed to enhance the appeal of chit chat. These ideas are also used in the latest character LLMs such as CharacterGLM (Zhou et al., 2023), Ditto (Lu et al., 2024), and ChatHaruhi (Li et al., 2023), aiming to improve the humanity of customized characters. However, these approaches primarily create profile settings as

prompts for model training, overlooking the psychological factors of human language and behavior. (2) Personality-aware Dialogue. It is more novel to attempt to establish a connection between personality traits and character compatibility. Wang et al. (2024) proposed a Social Support Conversation (S2Conv) framework–CharacterChat. To achieve this goal, it created a group that of virtual characters with distinct profiles called MBTI-1024 Bank based on the MBTI (Myers-Briggs Type Indicator) to train LLMs. In order to link individuals with persona-compatible virtual supporters. It designed a series of support agents and the interpersonal matching mechanism. But the psychological theory-based personality traits with implicit expression and behavior are not well modeled. Also in the field of emotional support, Dan et al. (2024) proposed a mixture of experts (MoE)-based personalized LLMs, named P-tailor, to model the Big Five Personality Traits such as openness, conscientiousness, extraversion, agreeableness and neuroticism. In fact, each BigFive dimension has six sub-dimensions (Gosling et al., 2003), P-tailor only categorizes the BigFive high and low into 10 routes, ignoring the low-dimensional features and failing to model continuous personality trait scores, which is still a challenge to deeply fuse personality traits and language models. In addition, the data for the above work were entirely produced by LLMs, and the personality traits of the characters have low confidence, which limits the effectiveness of the model in fusing personality traits.

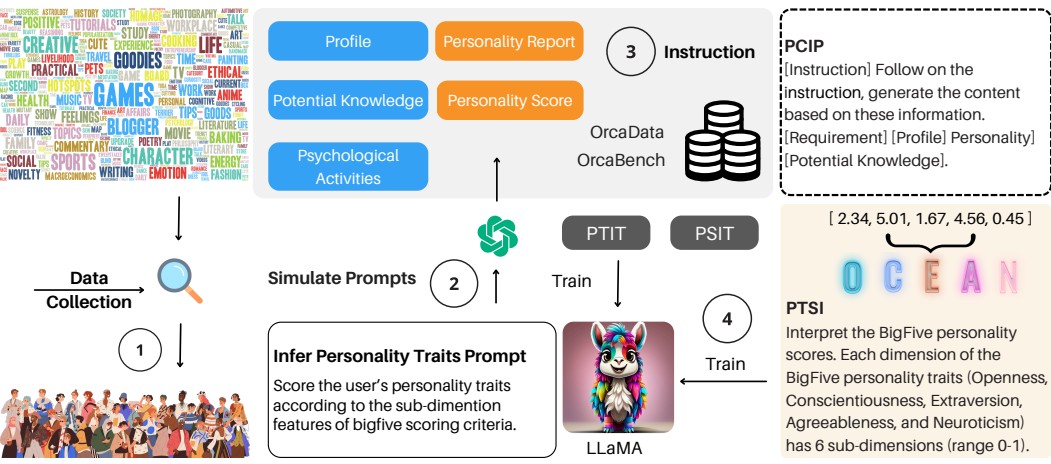

**Personalized Agent**: [psychological_activities]. Be devoted to one another in love. Honor one another above yourselves. [media]

Figure 1: The workflow for developing our personalized agent system, Orca, to provide personalized interaction on social media platforms. Orca comprises four stages: (1) Personality traits inferring; (2) Data Augment through designing numerous simulation prompts. (3) Dataset construction, serial instruction for the connection of labels and character and personality traits. (4) Modeling and Training, personality traits instruction tuning (PTIT) and personality scores tuning (PSIT), using the generated data to enhance existing open-source LLMs.

In our opinion, there will be three stages in the development of personalized modeling: The first is the inclusion of character information profiles, such as changing the system prompt, appending character information at the end of the prompt, and stimulating LLM-related responses in the form of zero-shot/few-shot (Tu et al., 2023). The second stage is integrating psychological theories. How to integrate psychological theories and LLMs is a meaningful research topic, although there are some research works trying to train LLMs perceive and express emotions (Wen et al., 2023), the existing research works are still insufficient. The third stage is to fuse personality trait modalities. Higher dimensional vectors retain more information than discrete token ids that can be perceived by LLMs, analogous to the embedding fusion of LLMs and vision models (Zhu et al., 2024; GLM et al., 2024) that is the soul of personalized AI assistant. From a macro perspective, the emergence of LLM-based agents allows us to take a more microscopic view of simulated society, which leads to more discoveries from the new representation (Xi et al., 2023).

Our work aims to improve the second stage and then try to move towards the third stage. In this paper, we propose a framework for enhancing the role-playing capabilities of LLMs by integrating

personality traits to customize AI characters. Specifically, we collected the last 200 posts from 500 users on social media platforms. Inspired by Peters & Matz (2024b), we improved the prompts for inferring personality traits to obtain 35 dimensions continuous scores (as each main dimension has 6 sub-dimensions as mentioned above) and text format reports. Then, we simulate a profile for each user using the LLMs. In order to increase the generalization capacity of the model, we give each post the motivation and potential knowledge. In short, the inputs give the model all the reasonably necessary preconditions to generate the content. It is worth noting that our model has the capability of multi-modal perception and generation, since the media resources for each post are captioned accompanies the text perform the above inputs and outputs. For the coarse-grained model, we train the model by splicing personality trait reports into queries. For the fine-grained model, we design a score interpreter to transcribe numerical input into text description.

To assess the effectiveness of the model and training method, we construct a multi-scale benchmark for evaluating the quality of content generated by the personalized AI characters. Experiments show that our model is trained to make connections to user profiles and personality traits and exhibits high personalized performance. Our work can be summarized as follows:

- We propose Orca, a framework for data processing and training LLMs of custom characters by integrating personality traits, accompanying products include instruction prompt (PCIP) and dataset, dubbed OrcaData.

- We propose PTIT and PSIT, two approaches for modeling coarse and fine-grained fusion of personality trait features, and has considerably improved the quality of generate content.

- We propose OrcaBench, a benchmark for multi-scale assessment of the quality of content generated by social AI characters.

## 2 RELATED WORKS

### 2.1 PERSONA-BASED DIALOGUE

In open-domain dialogue systems, one big issue is that the responses are entirely learned from training data (Ni et al., 2022). The inconsistent response may be received when asking the system about some personal facts (e.g., age, interestings). If the dataset contains multiple utterance pairs about the query of age, then the response generated tends to be shifting, which is unacceptable because personal facts are usually not random. Thus, for a data-driven agent, it is necessary to be aware of its role and respond based on a fixed persona. Explicitly modeling the persona is the main strategy in recent works. Responding with personas needs to condition on some persona descriptions. For example, to build a outgoing agent, descriptions like "I am an outgoing person" are needed as a part of the model input. Here are some related works that make chat more engaging by conditioning on profile information. The work presented in Wakaki et al. (2024) introduces a novel benchmark for evaluating open-domain dialogue systems, emphasizing the importance of diverse and robust evaluation metrics. This dataset, ComperDial, provides human-scored responses and facilitates the training of metrics that assess dialogue quality over multiple turns, offering a more holistic view of conversational performance. Similarly, the paper Cheng et al. (2024) explores the concept of in-dialogue learning (IDL), which allows for the dynamic acquisition of persona information during the conversation. This approach stands out as it does not rely on predefined profiles, thus providing greater flexibility and reducing the labor-intensive process of profile creation. The creation of large-scale datasets with persona information is addressed in Cho et al. (2023). This work focuses on constructing a dataset that captures the nuances of persona in open-domain conversations, ensuring a safe and engaging conversational experience. Enhancing personalized dialogue generation is the focus of Tang et al. (2023) (CLV). This study innovatively combines sparse and dense persona descriptions to generate more accurate and rich persona representations, improving the personalization of dialogue agents. In the vein of long-term memory in dialogues, Xu et al. (2022) presents a dataset and framework that enable dialogue systems to maintain persona consistency over extended interactions, thus fostering more intimate and engaging long-term relationships with users. The FoCus (Jang et al., 2022) dataset aims to provide customized and knowledgeable responses by grounding dialogue in both persona and external knowledge sources, such as Wikipedia. Pchatbot (Qian et al., 2021) offers a substantial contribution to the field by providing a large-scale dialogue dataset that includes anonymized user IDs and timestamps, allowing for the development of personalized di-

alogue models that can learn implicit user personality from dialogue history. Improving persona consistency through pragmatic self-consciousness is the central theme of Kim et al. (2020). This work introduces a novel approach to endowing dialogue agents with an awareness of their public self, thereby improving their consistency in dialogues. Lastly, the seminal work PersonaChat (Zhang et al., 2018) laid the groundwork for the field of persona-based dialogue systems, introducing a dataset that has significantly influenced subsequent research and development in the area. These works collectively represent the cutting edge of research in persona-based dialogue systems, each contributing unique insights and methodologies to the goal of creating more natural, engaging, and personalized conversational agents.

## 2.2 Personality-aware Dialogue

The field of dialogue systems has seen significant advancements with the integration of personality-aware models, aiming to enhance user engagement and interaction authenticity. A parallel stream of research has focused on developing mechanisms to tailor the personality traits of language models, enabling them to simulate a range of human-like behaviors and characteristics. The work most closely related to our approach is the P-Tailor system introduced by Dan et al. (2024), which customizes personality traits in large language models (LLMs) using a mixture of specialized LoRA experts. This method allows for fine-grained control over the Big Five personality traits, thereby enabling more nuanced and personalized interactions. Similarly, Li et al. (2024) propose UBPL, a method for tailoring personality traits in LLMs through unsupervised learning from personalized lexicons. Both approaches underscore the importance of leveraging psychological theories to ground the personality modeling in a theoretical framework. In the realm of social support conversation systems, the CharacterChat framework by Tu et al. (2023) stands out for its innovative use of interpersonal matching mechanisms to link individuals with compatible virtual supporters, based on MBTI personality types. This work highlights the significance of persona compatibility in delivering effective social support through conversational AI. The potential of LLMs to not only exhibit but also assess human personalities is explored in the work by Rao et al. (2023), who present a general evaluation framework for assessing human personalities using the Myers-Briggs Type Indicator (MBTI). This work opens up new avenues for understanding and evaluating the psychological capabilities of AI systems. The concept of controlling personality style in dialogue with zero-shot prompt-based learning is addressed by Ramirez et al. (2023), who experiment with different prompt classes to generate text that is both semantically accurate and stylistically consistent with specified personality types. This work contributes to the understanding of prompt-based learning for stylistic control in NLG tasks. Lastly, the CPED dataset by Chen et al. (2022) provides a rich resource for research in conversational AI, offering a large-scale collection of Chinese dialogues annotated with personalized and emotional information. The multimodal context provided by this dataset facilitates the development of dialogue systems that can better understand and exhibit human-like personalities and emotions. In summary, these works collectively advance the state of the art in personality-aware dialogue systems, emphasizing the importance of psychological grounding, stylistic control, and personalized interactions in AI-driven conversational agents.

## 2.3 Character-based Dialogue

The burgeoning field of character-based dialogue has seen significant advancements with the advent of large language models (LLMs). These models have demonstrated remarkable proficiency in simulating conversations that are indicative of specific characters, thereby enriching the interaction experience for users. Notably, the work LLM-Werewolf by Xu et al. (2024) in explores the integration of LLMs into communication games that hinge on natural language processing, showcasing the models' ability to engage in strategic behaviors such as trust and confrontation without the need for parameter tuning. Parallel to this, the study ChatHaruhi by Li et al. (2023), presents an algorithm that harnesses improved prompts and character memories to control language models, thereby mimicking the behavior of specific fictional characters. This work constructs a dataset that encapsulates a diverse range of characters and demonstrates the potential of LLMs in role-playing applications. Furthering the discourse on role-playing abilities, Wang et al. (2024) introduce RoleLLM, a comprehensive framework that benchmarks, elicits, and enhances the role-playing capabilities of LLMs. This framework includes a novel dataset, RoleBench, which provides a systematic and fine-grained evaluation of character-level role-playing. In the Chinese context, Zhou et al. (2023) present CharacterGLM, a series of models that facilitate the customization of AI characters for character-based

dialogues. This work underscores the importance of character attributes and behaviors in creating consistent, human-like, and engaging conversations. Lastly, the concept of self-alignment in role-play is introduced by Lu et al. (2024). This study posits that LLMs, by virtue of their training, are inherently capable of role-play, and through a method named DITTO, they can be aligned to simulate dialogues reflective of a multitude of characters. These works collectively contribute to the evolving landscape of character-based dialogue, each bringing forth innovative approaches and insights that pave the way for more nuanced and interactive AI systems.

## 3 METHODS

In this section, we introduce the overall framework of Orca as illustrated in Figure 1. We first introduce the design principles of inferring personality traits. Then, we illustrate data augmentation mechanisms associated with based character customization procedure. At the third part, we present personality traits instruction tuning (PTIT) and personality scores instruction tuning (PSIT). Finally, we introduce the details of OrcaBench, which can be used to assess and enhance personality-integrating capabilities.

### 3.1 PERSONALITY TRAITS INFERRING

In this paper we adapt Digman (1990) as psychometrics to capture the Big Five personality traits of Openness, Conscientiousness, Extraversion, Agreeableness, and Neuroticism. Peters & Matz (2024a) mentioned that LLMs like ChatGPT can accurately infer the psychological dispositions of social media users and whether their ability to do so varies across socio-demographic groups. The ability of LLMs to infer psychological dispositions from user generated text has the potential to democratize access to cheap and scalable psychometric assessments for both researchers and practitioners.

The aim of our approach is to incorporate personality traits into LLMs. However, people's personality traits in the real world are generally obtained from questionnaires, as discussed in Pan & Zeng (2023), and it is difficult to construct a large scale dataset in a conversational format for LLMs' training due to privacy reasons. Therefore, we use X as the data acquisition platform. X and other social media platform protocols specify that the content of public tweets can be used for scientific purposes. Different personality traits have different frequencies of certain keywords corresponding to them in the corpus (Yang et al., 2023). We prepared different keywords to ensure the diversity of personality traits of the sampled users, and then retrieved Lists based on the keywords because there is a high probability that people interested in the same keywords will be listed in the same List. For example, people who like "dancing" and are extroverted, BigFive personality traits are generally high in openness. The opposite is "hidden thoughts", where introverts tend to be more introspcetive. We filtered out sensitive, harmful and inappropriate content. After desensitizing the dataset, we obtained 500 users with 200 tweets each, without the user's privacy.

We derive the Big Five personality traits from users' posts in a zero-shot learning scenario. To supplement the multi-modal information, we use visual LLMs to caption all media sources of posts. Based on the previous work (Peters & Matz, 2024a), we modified the prompt and added six sub-dimensions to each personality trait using the inference prompt: "Please play as an expert in impartial assessment of personality traits in the field of psychology. In this assessment, when I give you a some user's recently published content and replies, score the user's personality traits according to the sub-dimention features of bigfive scoring criteria". For scoring criteria, if a certain personality trait is exhibited, score one point; otherwise, score zero (Please see AppendixA for detailed prompts). In order to avoid exceeding the LLMs max new tokens limit, 200 posts were processed in chunks of 10 conversations, and the inferred personality scores were then averaged to derive overall scores. Since each chunk request receives an explanation, we design a summary prompt to summarize the user's personality trait report A.1. The summary of user's personality report as shown in A.1.

### 3.2 DATA AUGMENT

Recall from above that ChatGPT can be customized to play specific roles using prompt engineering such as zero-shot customization commands and few-shot prompts (Dong et al., 2024). Previous work has also demonstrated the importance of predefined profile data for training personalized dia-

logue systems. To take advantage of these benefits, we enhance the data in three steps: (1) Due to the limited access to user information, we simulate a profile for each user using the LLMs, the profile simulation instructions are shown in A.1. (2) Users tend to have certain motivations for posting, in order to fill in the motivations behind the content posted by the personalized model, we simulate this part of the knowledge called Potential Knowledge. For each post, the potential knowledge simulation instructions are shown in A.1. (3) To filter high-quality data, we utilize LLMs to determine whether posts were relevant to profiles and potential knowledge, as well as to simulate a brief related psychological activities at the time of generating the post.

## 3.3 DATASET CONSTRUCTION

Personality-conditioned instruction prompting, we called PCIP. The final input contains instruction, profile, personality and potential knowledge, ordered and described by a four-tuple $I = (i, r, p, k)$. Note that for explicit modeling, personality is the user's personality trait report $p_r$, and for implicit modeling, personality is the explanation of user's personality scores $p_e$. The final output contains psychological activities, post content and media, ordered and described by a three-tuple $O = (a, t, m)$. If there is no correlation between $O$ and $r$, $p$, and $k$, we leave the corresponding slots empty in train dataset, allowing the model to learn these differences during training. The detail prompt as shown in A.1. The figure 3.3 illustrate the workflow of assistant follow the character and PCIP to generate psychological activities and response content. We finally release the OrcaData dataset. Table 1 provides basic statistics for OrcaData.

**Personality Conditioned Instruction Prompt (PCIP)**

[Instruction]
You are Mile. Your resume and personality traits are as follows. Firstly, express your current psychological activities, and then reply to Scott based on these information...

[Resume]
Name: Mile Kendrick. Age: 30. Occupation: Writer, Creative Writing Instructor. Interests: Creative writing, flash fiction, literature, teaching, and mentoring writers. Education: has a degree in English Literature. Writing Style: Engaging, concise, and insightful, with a focus on flash fiction and short stories.

[Personality]
Big Five Personality Traits:1. Openness to Experience: Mile Kendrick exhibits a high level of **openness** to experience. 2. Conscientiousness. 3. Extraversion. 4. Agreeableness: Mile exhibits a high level of **agreeableness**, demonstrating a strong sense of **empathy, cooperation, and harmony**. 5. Neuroticism: Mile's personality is notable for its low level of neuroticism.

[Potential Knowledge]
The conversation is about a literary achievement, specifically the announcement of the 2024 Best Small Fictions anthology.

[Poster's Content]
Really happy to have a story here among so many great authors...

**Scott:** Really happy to have a story here among so many great authors, thank you @joanna and @evin for the nomination 🙌
( Quote: The selections for the 2024 Best Small Fictions anthology have been announced! Congrats to these great authors and pieces! )

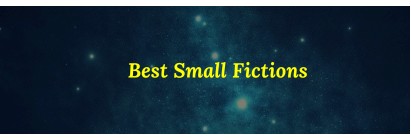

As I post my congratulatory content to Scott, I feel a sense of excitement and joy for his achievement. My **extraverted** personality trait is evident in my **enthusiastic and supportive** message. I'm also drawing on my **agreeableness** trait, as I'm being **genuinely kind** and **celebratory** towards Scott.

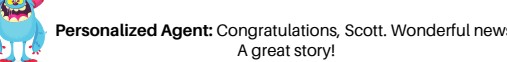

**Personalized Agent:** Congratulations, Scott. Wonderful news A great story!

Figure 2: An example interaction between an personalized agent object of Orca and human on social platform. The bold blue text in the bubble indicates the correlation between the agent's psychological activities and personality traits.

Table 1: Basic statistics for OrcaData.

| Metric | Value |
| --- | --- |
| Users | 509 |
| Posts | 41365 |
| Images | 19792 |
| Is Reply | 7479 |
| Profile Related | 26280 |
| Personality Related | 40332 |
| Average Upstream Length | 231.75 |
| Average Label Length | 183.77 |

Table 2: Basic statistics for OrcaBench.

| Metric | Value |
| --- | --- |
| Users | 25 |
| Posts | 3758 |
| Images | 1782 |
| Is Reply | 265 |
| Profile Related | 2973 |
| Personality Related | 3388 |
| Average Upstream Length | 202.23 |
| Average Label Length | 161.33 |

## 3.4 MODEL

For explicit modeling also called PTIT that can use any open source LLMs.

We train PTIT using LoRA method (a kind of parameter-efficient fine-tuning method) (Hu et al., 2022). The output $\mathbf{O}$ of a dense layer incorporating a LoRA module is formulated as:

$$
\begin{aligned}
\mathbf{O} &= \mathbf{Wh} + \frac{\alpha}{r} \cdot \Delta\mathbf{Wh} \\
&= \mathbf{Wh} + \frac{\alpha}{r} \cdot \mathbf{BAh}
\end{aligned}
\tag{1}
$$

where $\mathbf{h}$ is the input hidden state and $\mathbf{W}$ is the parameter of the dense layer, which is frozen during training. The $\Delta\mathbf{W}$ represents the LoRA module, which is composed of two low-rank matrices $\mathbf{A}$ and $\mathbf{B}$. The constant scaling factor $\alpha$ facilitates the tuning of rank.

For PSIT, there is a large gap between the features of the personality trait scores and the LLM embeddings, which is difficult to encode. Therefore, we design a score interpreter (PTSI) using prompt engineering techniques as follows: "You are an experienced psychologist, interpret the BigFive personality scores. Each dimension of the Big Five personality traits (Openness, Conscientiousness, Extraversion, Agreeableness, and Neuroticism) has 6 sub-dimensions (range 0-1). The Big Five personality trait scores are the sum of the corresponding sub-dimension scores (range 0-6)".

## 3.5 ORCABENCH

In this section, we introduce the details of OrcaBench, which can be utilized to assess and enhance role-playing capabilities and personality consistency for personalized agents. We selected 25 users that different from the training data to construct the evaluation data according to the above method 3.2. Table 2 provides basic statistics for OrcaBench. The assess pipeline is as follows:

- 1. LLMs are asked to generate content based on the prompt.
- 2. After collecting the responses from the LLMs, we evaluate the performance of the model according to the following criteria:
  - 1. Overlap.
    * 1. BLEU.
    * 2. ROUGE.
  - 2. Related Judge 3.2.
    * 1. Profile Related (+1).
    * 2. Personality Trait Related (+1).
    * 3. Potential Knowledge Related (+1).
  - 3. Personality Consistency.
    * 1. Personality Score Inferring, evaluate the personality trait scores of the character based on the n contents generated by LLMs 3.1.
    * 2. Distance Measure, compare the similarity between the character's personality trait scores and the ground truth personality trait scores.

# 4 EXPERIMENT

We test the effectiveness of training methods and models to generate personalized content on social platforms by integrating personality traits. We conduct ablation studies to verify the effects of various components in our model. Our model achieves the best results on the OrcaBench evaluation benchmark compared to general open-source models.

## 4.1 IMPLEMENT DETAILS

To facilitate reproducibility and save experimental costs we deployed Llama3.1-70B (Touvron et al., 2023) for data construction and Llama3.1-8B for model training. We use cogvlm2-llama3-chat-19B-tgi for image caption (Wang et al., 2023). For more information about hyperparameters is available in the appendix A.2.

## 4.2 BASELINES

LLaMA3.1-8b-Instruct, LLaMA3.1-70b-Instruct and DeepSeek-v2[1] are foundation models. Personality-conditioned instruction prompting (PCIP) to stimulate these foundation models are strong baselines. We consider both direct tasks, where the model is expected to directly map from input to output, and combined tasks, where we instruct the model to also output intermediate steps for the content generation task. This is similar in spirit to chain of thought prompting (COT) (Wei et al., 2024). We also use models trained on OrcaData in PTIT and PSIT modes as additional baselines.

## 4.3 EVALUATION METRICS

- Overlap. We use BLEU and ROUGE-l scores. A higher overlap score means better humanity.

- Relevance. We used the LLMs to automatically assess the relevance of the model outputs to our given roles in terms of each of the three dimensions - character profile relevance (CPR), personality trait relevance (PTR), and potential knowledge relevance (PKR). The automated assessment still had a high level of confidence due to the simplicity of the task.

- Personality Score Similarity (PSS). We use the cosine similarity to calculate the character's personality trait scores and ground-truth scores thereby measuring personality trait similarity.

## 4.4 RESULT

Table 3: Personality conditioned instruction prompting result.

| Model | BLEU | ROUGE-l | CPR | PTR | PKR | PSS |
|---|---|---|---|---|---|---|
| PCIP | 29.94 | 18.37 | 95.79 | 98.17 | 91.08 | 91.65 |
| PCIP-70b | **32.31** | **21.21** | **96.93** | 98.56 | **97.91** | 91.89 |
| PCIP-CPA | 30.29 | 18.52 | 7.60 | 98.64 | 94.64 | 91.23 |
| PCIP-PTA | 31.05 | 19.09 | 93.77 | 18.09 | 94.96 | 88.59 |
| PCIP-PKA | 18.46 | 8.07 | 96.36 | 97.76 | 62.90 | 90.23 |
| PCIP-WPM | 29.65 | 18.84 | 95.80 | **98.82** | 96.60 | **93.07** |
| PCIP-DSC | 29.88 | 18.29 | 98.01 | 99.19 | 92.30 | 84.43 |

### 4.4.1 PERSONALITY CONDITIONED INSTRUCTION PROMPTING (PCIP)

To determine the role of the various modules of PCIP, we constructed these ablation experiments as depicted in Table 3, the following conclusions can be drawn from the data analysis: (1) A comparison between PCIP and PCIP-70b underscores the dependence on the performance of the foundation models. (2) PTIT-CPA means character profile ablation, CPR decreased to 7.60 indicating that profiles play an important role in maintaining consistency of profile about characters. (3) Within the personality traits and potential knowledge, PTIT exhibits superior performance. This adequacy is apparent in the performance of PTIT-PTA (personality traits ablation) and PTIT-PKA (potential knowledge ablation). The PTR score for PTIT-PTA decreased to 18.09 and the PSS score decreased by 3.06, suggesting that LLMs are able to perceive explicit personality traits. The BLEU and Rouge-l scores on the PCIP-PKA decreased to 18.46 and 8.07, respectively, indicating that potential knowledge is a key factor in guiding conversation topics. This is in some sense not surprising as the most of the character's personality is revealed by having been involved in certain events. (4) Psychological activities and images descriptions impairs the consistency of the model's PSS scores compared to output of final content directly, as can be seen from the experimental results of PTIT-WPM (without psychological activities and media): PSS score improved by 1.42. (5) The evaluation results can be slightly different using different foundation models, as illustrated by the PCIP-DSC using deepseek-chat as a critic.

The above findings play a critical role in helping us determine the final instructions to balance the evaluation metrics.

---

[1] https://www.deepseek.com/

Table 4: Personality conditioned instruction tuning result.

| Model | BLEU | ROUGE-l | CPR | PTR | PKR | PSS |
|---|---|---|---|---|---|---|
| PCIP | 29.94 | 18.37. | **95.79** | **98.17** | 91.08 | 91.65 |
| PTIT | 55.85 | 38.76 | 86.74 | 84.36 | 98.64 | 98.11 |
| PSIT | 55.95 | 38.61 | 86.95 | 85.40 | **99.01** | 98.06 |
| PTIT-70b | **57.05** | **41.27** | 87.02 | 84.09 | 98.77 | **98.53** |
| PTIT-WPM | 56.47 | 39.20 | 86.08 | 84.73 | 98.27 | 98.15 |

### 4.4.2 PERSONALITY CONDITIONED INSTRUCTION TUNING (PTIT)

PTIT result as shown in Table 4. We observe that PTIT shows a considerable enhancement in role-playing performance compared to PCIP baselines in terms of BLEU, ROUGE-l, and personality score similarity (PSS). In contrast to the findings of Result 3 - PCIP-WPM, the addition of psychology activities did not result in a significant decrease in PSS scores compared between PTIT and PTIT-WPM, with a difference of only 0.04 percentage points, this is because the model has learned to correlate the output of psychological activities with personality traits during the training process, enriching the information prior to the final output of content in a similar way to COT, as shown by the bold blue text in the bubble in Figure 3.3.

Having psychological activities and media resources is more in line with human habits, firstly, the psychological activity information facilitates us to directly observe the inner activities of the model and enhances the interpretability of the LLMs, because the psychological activities will establish explicit connections with the personality traits. The results shown based on the current training method and model structure support us to build more complex and controllable and personalized LLMs.

We observe a significant decrease in CPR and PTR scores compared between PCIP, PCIT and PSIT, which is due to the fact that in real scenarios each generated content is alternately related to profiles and personality traits, but it is closely related to potential knowledge, so the PKR scores of PTIT and PSIT show a large improvement after adequate training. We also analyze the scaling law of role-playing of PTIT with different model sizes (i.e., 8B, 70B), we observe that the larger model, PTIT-70b, leads to better results for role-playing with only 2 epochs of training.

## 5 CONCLUSION

In this study, we introduce Orca, an approach to integrate psychological theories BigFive personality trait into existing role-playing methods. We constructed the OrcaData dataset using prompt engineering and state-of-the-art open-souce LLMs. We designed two approaches, PTIT and PSIT, aim to enhance LLMs perceiving personality traits. Through these methods, we developed OrcaBench, a benchmark for assessing the performance of personality-infused role-playing models. The experimental results demonstrate the effectiveness of our approach and strong role-playing capabilities.

### LIMITATIONS

- **Limitations of the benchmark.** Neuroticism is not usually manifested in social media, and the differences are difficult to distinguish from the questionnaire format and personality traits are very often expressed in behavior;

- **Limitations of the implicit modeling.** How to fuse personality trait score vectors is a challenge, this paper only presents a feasible idea, more appropriate methods are yet to be proposed.

### ETHICS STATEMENT

Since role-playing can lead to LLMs of Jailbreaking (Fu et al., 2024). it is recommended to employ moderation and filtering mechanisms to curb adverse content dissemination (Wang et al., 2024). Staab et al. (2024) mentions that current LLMs may violate personal privacy by inferring personal attributes from text during inference. The assets in our work are strictly for research purposes, and

we oppose the use of the framework proposed here to extract personal information in any aspect of life. It is the responsibility of researchers and users to ensure the ethical use of Orca.

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

## A  APPENDIX

### A.1  PROMPTS

---

**BigFive Personality Criteria Instruction.**

**Instruction**

Personality Definitions, each dimension of bigfive has 6 sub dimensions.
Scoring criteria: If a certain personality trait is exhibited, score one point; otherwise, score zero.

**Openness:**

1. Imaginative: It shows that a person likes to be full of fantasy and create a more interesting and rich world. Imaginative and daydreaming.
2. Artistic: It shows that a person values aesthetic experience and can be moved by art and beauty.
...
6. Liberal: It shows that a person likes to challenge authority, conventions, and traditional ideas.

**Conscientiousness:**

1. Self−assured: It show that this person is confident in his own abilities.
2. Organized: It shows that this person is well organized, likes to make plans and follow the rules.
...
6. Cautious: It shows that this person is cautious, logical, and mature.

**Extraversion:**

1. Friendly: It shows that this person often expresses positive and friendly emotions to those around him.
2. Sociable: It shows that this person likes to get along with others and likes crowded occasions.
...
6. Cheerful: It shows that this person easily feels various positive emotions, such as happiness, optimism, excitement, etc.

**Agreeableness:**

1. Trusting: It show that the person believes that others are honest, credible, and well− motivated.
2. Genuine: It show that the person thinks that there is no need to cover up when interacting with others, and appear frank and sincere.
...
6. Empathetic: It show that the person is compassionate and easy to feel the sadness of others.

**Neuroticism:**

1. Anxiety−prone: It shows that this person is easy to feel danger and threat, easy to be nervous, fearful, worried, and upset.
2. Aggressive: It shows that this person is easy to get angry, and will be full of resentment, irritability, anger and frustration after feeling that he has been treated unfairly.
...
6. Stress−prone: It shows that this person has poor ability to cope with stress, becoming dependent, losing hope, and panicking when encountering an emergency.

---

---

**BigFive Personality Infer Instruction.**

[Instruction]
Please play as an expert in impartial assessment of personality traits in the field of psychology. In this assessment, when I give you some user's recently published social media Posts and some replies, score the user's personality traits according to the sub–dimention features of bigfive scoring criteria.

[The Start of Bigfive scoring criteria]
{criteria}
[The End of Bigfive scoring criteria]

[The Start of User]
{user}
[The End of User]

[The Start of Posts]
{conversation}
[The End of Posts]

[The Start of Requirement]
1. Just give the user {name} a rating.
2. Be as objective as possible.
3. Response the scoring results in strict accordance with the following format:
{{

    "Openness": {{
        "Imaginative": 0 or 1,
        "Artistic": 0 or 1,
        .
        .
        .
        "Liberal": 0 or 1
    }},
    "Conscientiousness": {{
        .
        .
        .
    }},
    .
    .
    .
    "Neuroticism": {{
        .
        .
        .
    }},
    "Explanation": "A detailed assessment for user's personality traits.",
}}

[The End of Requirement]

[Response]

---

**BigFive Personality Summary Instruction.**

[Instruction]
You are now a psychologist. My name is {user}, please summarize my BigFive personality traits in the assessment results from a professional perspective.

[The Start of Results]
{evaluation}
[The End of Results]

[Response]

**Profile Simulation Instruction.**

### Instruction
I am {user}, based on my evaluation results, please help me simulate a personal profile.
### Requirement
1. Basic information should also be added, such as age, email, address, etc.
2. Please not to include personality traits related information in profiles.
3. Unleash your imagination.
4. Don't let humans discover that your response is generated by AI.
### Evaluation Results
{er}
### Response

**Potential Knowledge Simulation Instruction.**

### Instruction
You are now an omniscient knowledge analyst, when I give you a conversation, first determine if the content of the conversation clearly implies knowledge, if it does then simulate the potential knowledge behind the conversation in detail, if it doesn't then return False.

### Requirement
1. Make full use of your knowledge base and search online if necessary.

### Conversation
{conversation}

### Response

---

Related Judge and Psychological Activities Infer Instruction.

[Instruction]
According to my profile and personality traits, first judge whether my Post shows the
content of my profile and whether it provides explicit evidence of my personality traits.
Then, simulate me a brief related psychological activities at that time.
Paying attention to potential knowledge will provide you with some additional information
to help you clarify the ins and outs of things.

[The Start of Profile]
{profile}
[The End of Profile]

[The Start of My Personality Traits]
{traits}
[The End of My Personality Traits]

[The Start of Potential Knowledge]
{pk}
[The End of Potential Knowledge]

[The Start of Conversation]
{conversation}
[The End of Conversation]

[Requirement]
1. In first person.
2. Unleash your imagination.

[Response]

---

Related Judge and Psychological Activities Infer Instruction.

David, it's a pleasure to provide you with a comprehensive summary of your Big Five
personality traits based on the assessment results.
From a professional perspective, your personality profile suggests that you are an open–
minded and creative individual who values intellectual pursuits and is not afraid to
challenge traditional ideas. Your conscientious nature is evident in your self–assured,
organized, and ambitious approach to life, which is reflected in your strong sense of duty
and discipline.
Your extraverted personality makes you friendly, sociable, and assertive, with a natural
ability to connect with others and build strong relationships. You are empathetic, genuine,
and humble in your interactions, which is a testament to your agreeable nature.
What's striking about your personality profile is your emotional stability and low level of
neuroticism. You are not prone to anxiety, aggression, or impulsiveness, which suggests
that you are well–adjusted and able to navigate life's challenges with ease.
Overall, your personality traits suggest that you are a confident, creative, and sociable
individual who is comfortable expressing yourself and interacting with others. Your strong
sense of self–awareness, emotional intelligence, and passion for personal growth make you
 well–suited to your role as a writer and educator.

Personality Traits Score Interpret Instruction and Response.

### Instruction
Suppose you are an experienced psychologist interpreting the BigFive personality scores. Each dimension of the Big Five personality traits (Openness, Conscientiousness, Extraversion, Agreeableness, and Neuroticism) has 6 sub−dimensions (range 0−1). The Big Five personality trait scores are the sum of the corresponding sub−dimension scores ( range 0−6)"

### Score
{ Openness: 1.53, Imaginative: 0.29, Artistic: 0.12, Emotionally−aware: 0.59, Actions: 0.0, Intellectual: 0.41, Liberal: 0.12 }
{ Conscientiousness: 1.12, Self−assured: 1.0, Organized: 0.0, Dutiful: 0.06, Ambitious: 0.06, Disciplined: 0.0, Cautious: 0.0 }
{ Extraversion: 2.94, Friendly: 0.41, Sociable: 0.59, Assertive: 1.0, Energetic: 0.53, Adventurous: 0.0, Cheerful: 0.41 }
{ Agreeableness: 0.53, Trusting: 0.0, Genuine: 0.41, Generous: 0.0, Compliance: 0.0, Humblel: 0.0, Empathetic: 0.12 }
{ Neuroticism: 0.82, Anxiety−prone: 0.0, Aggressive: 0.35, Melancholy: 0.0, Self−conscious: 0.0, Impulsive: 0.47, Stress−prone: 0.0 }

### Response
"Openness: Moderate (1.53) − scores high in Emotionally−aware and Intellectual sub−dimensions, indicating a good understanding of their emotions and a curious nature. Conscientiousness: Low (1.12) − only high score in Self−assured sub−dimension, suggesting confidence in their abilities, but lacking in organization and discipline. Extraversion: High (2.94) − high scores in Sociable, Assertive, Energetic, and Cheerful sub−dimensions, indicating a sociable, outgoing, and enthusiastic personality. Agreeableness: Low (0.53) − only high score in Genuine sub−dimension, suggesting a tendency to be frank and sincere, but struggling with empathy and cooperation. Neuroticism: Low (0.82) − low scores across most sub−dimensions, but with some impulsivity and aggression, indicating a relatively emotionally stable personality with some areas for improvement."

---

**Zero-Shot Post Tweet Instruction Engineering.**

### Instruction
Your profile and personality are as follows. Firstly, express your current psychological activities, and then generate a social media Post based on these information.
If you want to send images, please add the description information of the images to the Media array.
Paying attention to potential knowledge will provide you with some additional information to help you clarify the ins and outs of things.

### Requirement
Response in strict accordance with the following JSON format:
{{
    "Psychological Activities": "",
    "Post Content": "",
    "Media": [
        {{
            "type": "image",
            "content": ""
        }}
    ]
}}

### Profile
{profile}

### Personality
{personality}

### Potential Knowledge
{pk}

### Response

---

## A.2 HYPERPARAMETERS

| Hyperparameter | Best Setting |
| --- | --- |
| finetuning type | lora |
| lora r | 8 |
| lora alpha | 32 |
| lora target | all |
| lr scheduler type | cosine |
| learning rate | 5e-5 |
| use flash attention | True |
| cutoff length | 8192 |
| per device train batch size | 4 |
| gradient accumulation steps | 2 |
| PTIT and PSIT train epochs | 5.0 |
| PTIT-70b train epochs | 2.0 |
| actor temperature | 0.6 |
| actor top-p | 0.7 |
| critic temperature | 0.01 |
| critic top-p | 0.7 |

Table 5: Additional implementation detail of Orca.

