# OpenReview forum: "Orca: Enhancing Role-Playing Abilities of Large Language Models by Integrating Personality Traits"
_ICLR.cc/2025/Conference — ICLR 2025 Conference Withdrawn Submission_

### Official Review · Reviewer_3mAg · 2024-10-17

**Soundness:** 1
**Presentation:** 2
**Contribution:** 2
**Rating:** 3
**Confidence:** 4

**Summary:**

The paper introduces Orca, a framework designed to enhance the role-playing abilities of large language models (LLMs) by integrating personality traits based on psychological theories, specifically the Big Five personality dimensions. Orca operates in four stages: (1) inferring users' personality traits from social media content using LLMs; (2) augmenting data by simulating user profiles, background stories, and psychological activities; (3) constructing datasets with personality-conditioned instruction prompts to stimulate LLMs; and (4) modeling and training LLMs through personality-conditioned instruction tuning methods (PTIT and PSIT). The authors also present OrcaBench, the first benchmark for evaluating the quality of LLM-generated content on social platforms. Experimental results demonstrate that Orca effectively incorporates personality traits into LLMs, significantly improving their role-playing capabilities and enabling the creation of more personalized and engaging conversational agents.

**Strengths:**

- The development of OrcaData provides valuable resources for training and evaluating personalized AI characters, facilitating future research in personalized dialogue systems. The work provides a reasonably large dataset
- The work is one of the first work tackling LLM persona performance on social media platform.

**Weaknesses:**

- The evaluation metric seems overfitting. To my personal understanding- the current ability of roleplay agent does not match the score provided in your benchmark- namely many of the model scoring unreasonably high. I suspect it might the judgement bias from the evaluator model
- Limited Psychological Scales tested. The work should also test on scales like MBTI to prove the effectiveness of their approach
- No human Evaluation has been performed.
- The benchmark is not talored towards social media content
- Lacking reference to related work
  - InCharacter: Evaluating Personality Fidelity in Role-Playing Agents through Psychological Interviews. In Proceedings of the 62nd Annual Meeting of the Association for Computational Linguistics (Volume 1: Long Papers), pages 1840–1873, Bangkok, Thailand. Association for Computational Linguistics.
  - On the Humanity of Conversational AI: Evaluating the Psychological Portrayal of LLMs ICLR 2024

**Questions:**

- How do you ensure soundness of your LLM based evaluations without comparing with human evaluation?
- Why did you only pick Big Five?
- Does the high score on OrcaBench reflect high performance as roleplay models?
- What additional test would you add to make the benchmark more oriented towards social media than a generic roleplay benchmark

---

### Official Review · Reviewer_qyKc · 2024-11-03

**Soundness:** 2
**Presentation:** 2
**Contribution:** 2
**Rating:** 3
**Confidence:** 3

**Summary:**

The paper proposes a framework called ORCA to improve the role-playing capabilities of large language models (LLMs) by incorporating personality traits based on the Big Five personality dimensions. The framework consists of four main stages: inferring personality traits, augmenting data with user profiles and simulated psychological activities, constructing a personality-conditioned dataset (OrcaData), and training the models using personality-conditioned instruction tuning (PCIP, PTIT, and PSIT). ORCA also introduces OrcaBench, a benchmark designed to evaluate the quality of content generated by these enhanced role-playing LLMs. Experimental results reportedly demonstrate that ORCA achieves improvements in generating personalized content and maintaining character consistency across various social interactions.

**Strengths:**

The paper investigates an important question, attempting to incorporate Big Five personality traits to enhance LLM role-playing abilities. The quality of the work is supported by a structured, multi-stage framework (ORCA) and a dedicated benchmark (OrcaBench) for evaluation.

**Weaknesses:**

I have concerns about the paper's method of constructing the dataset, as it is the central contribution of the paper, but is not clearly documented in the main paper. Specifically, the process for inferring personality traits from social media posts and constructing personality-conditioned prompts is not thoroughly detailed. This leaves questions about the reliability and reproducibility of the dataset. Additionally, the authors mention filtering and augmentation steps for data but do not explain these criteria, which are essential to understanding the data quality. Clearer documentation of these steps would strengthen the work by ensuring the dataset’s transparency and usability for future research.

**Questions:**

In terms of inferring the personality traits of the X posts, do you use human annotators or just an automated classifier? How would you justify the quality of your dataset?
Specifically, what model or technique was used if automated, or how many annotators and what guidelines were used if human-annotated?

---

### Official Review · Reviewer_RYfX · 2024-11-04

**Soundness:** 3
**Presentation:** 2
**Contribution:** 3
**Rating:** 3
**Confidence:** 4

**Summary:**

This paper presents Orca to incorporate personality traits. Orca consists of four stages: inferring personality traits, augmenting data, constructing datasets, and modeling and training. The paper also introduces OrcaBench, a benchmark for evaluating LLM-generated content on social platforms. Experiments show that Orca achieves better performance on OrcaBench.

**Strengths:**

1.The paper proposes a framework, Orca, that integrates personality traits into the data processing and training of custom LLM characters, addressing a gap in previous research.

2. The introduction of OrcaBench represents a significant contribution, providing a tool for evaluating the quality of LLM-generated content across multiple scales.

3. The experiments demonstrate Orca's superior performance on the OrcaBench  benchmark.

**Weaknesses:**

1.In terms of OrcaBench evaluation benchmark, the model is only compared to general open-source models. Experiments show that the proposed model achieves superior performance on this self-build benchmark. It is not enough to demonstrate its excellence of role-playing abilities. Maybe a human evaluation to assess the role-playing ability is more neutral.

2.BigFIVE (OCEAN) has its own questionnaire as a scale. Have you ever tried if your model role plays a certain personality and psychologists (or people with psychology background) conduct interviews, can it reflect a specific personality?

3.In this manuscript, there are several typos present, which significantly hinder the ease of reading. These mistakes can be particularly noticeable in certain paragraphs, such as the those included within the QUESTIONS. It appears that further refinement and careful proofreading are essential to enhance the overall readability and professionalism of the manuscript.

4.In Table 4, why does PTIT underperform PCIP with regards to CPR PTR PKR PSS by a large margin?

**Questions:**

1.In line 287-288, “ The figure 3.3 illustrate the workflow of assistant follow the character and PCIP to generate psychological activities and response content”, I have not found figure 3.3.

2.in line 449-450, “enriching the information prior to the final output of content in a similar way to COT, as shown by the bold blue text in the bubble in Figure 3.3.”,  I have not found Figure 3.3.

3.In line 445-446, “In contrast to the findings of Result 3 - PCIP-WPM, the addition of psychology activities did not result in a significant decrease in PSS scores compared between PTIT and”, what you do mean by “Result 3”. Maybe it’s the results in Table 3?

4.In Table 4, why does PTIT underperform PCIP regarding CPR PTR PKR PSS by a large margin?

5.In terms of OrcaBench evaluation benchmark, the model is only compared to general open-source models. Experiments show that the proposed model achieves superior performance on this self-build benchmark. It is not enough to demonstrate its excellence of role-playing abilities. Is there a human evaluation to assess the role-playing ability?

6.BigFIVE (OCEAN) has its own questionnaire as a scale. Have you ever tried if your model role plays a certain personality and psychologists (or people with a psychology background) conduct interviews, can it reflect a specific personality?

---

### Official Review · Reviewer_qyxa · 2024-11-06

**Soundness:** 1
**Presentation:** 2
**Contribution:** 1
**Rating:** 3
**Confidence:** 5

**Summary:**

This paper presents *Orca*, a framework for developing personalized dialogue systems by incorporating personality traits into large language model (LLM) training. Traditionally, most research on conversational agents has emphasized building character profiles to improve instruction-following. However, *Orca* addresses the psychological aspects of human interactions, integrating personality traits to create more nuanced and engaging role-play capabilities in dialogue systems. The *Orca* framework includes four key stages:

1. **Personality Traits Inferring**: Uses LLMs to analyze users’ Big Five personality traits, providing scores and personality profiles.
2. **Data Augmentation**: Simulates detailed user profiles, background stories, and psychological dynamics.
3. **Dataset Construction**: Implements personality-conditioned instruction prompting (PCIP) to generate personality-adapted dialogue data.
4. **Modeling and Training**: Applies personality-conditioned tuning methods (PTIT and PSIT) on open-source LLMs, using the tailored data to enhance role-playing abilities.

Additionally, the authors introduce *OrcaBench*, the first benchmark designed to assess LLM-generated content quality on social platforms with personality-driven factors. Experiments show that models trained under the *Orca* framework excel in recognizing and utilizing personality traits, resulting in enhanced and more authentic role-playing interactions.

**Strengths:**

This work merely presents a basic data processing method for role-playing conversation and lacks any significant contribution.

**Weaknesses:**

I suggest that the author give up rebuttals and instead spend more time considering interesting ideas.

There are a lot of similar works such as:

[1] RoleLLM: Benchmarking, Eliciting, and Enhancing Role-Playing Abilities of Large Language Models

[2] Character-LLM: A Trainable Agent for Role-Playing

[3] Large Language Models are Superpositions of All Characters: Attaining Arbitrary Role-play via Self-Alignment

**Questions:**

see weakness

---

### Note · Authors · 2024-11-13

I have read and agree with the venue's withdrawal policy on behalf of myself and my co-authors.